# Comparative Analysis of Bacterial Cellulose Membranes Synthesized by Chosen *Komagataeibacter* Strains and Their Application Potential

**DOI:** 10.3390/ijms23063391

**Published:** 2022-03-21

**Authors:** Monika Kaczmarek, Marzena Jędrzejczak-Krzepkowska, Karolina Ludwicka

**Affiliations:** Institute of Molecular and Industrial Biotechnology, Lodz University of Technology, 90-537 Lodz, Poland; marzena.jedrzejczak-krzepkowska@p.lodz.pl

**Keywords:** bacterial cellulose, *Komagataeibacter*, biomaterial, nanocomposite

## Abstract

This article presents a comparative analysis of bacterial cellulose membranes synthesized by several strains of the *Komagataeibacter* genus in terms of their specific physical, physico-chemical, and mechanical properties. Herein, the aim was to choose the most suitable microorganisms producing cellulosic materials with the greatest potential for the fabrication of bio-inspired nanocomposites. The selection was based on three main steps, starting from the evaluation of BNC biosynthetic efficiency with and without the addition of ethanol, followed by the assessment of mechanical breaking strength, and the physical parameters (compactness, structural integrity, appearance, and thickness) of the obtained biological materials. Ultimately, based on the performed screening procedure, three efficiently growing strains (*K. hansenii* H3 (6Et), *K. rhaeticus* K4 (8Et), and *Komagataeibacter* sp. isolated from balsamic vinegar (12Et)) were chosen for further modifications, enabling additional cellulose functionalization. Here, supplementation of the growth medium with five representative polymeric compounds (citrus/apple pectin, wheat starch, polyvinyl alcohol, polyethylene glycol) led to significant changes in BNC properties, especially dye loading abilities, mechanical strength, and water adsorption/retention capacities. The resulting nanocomposites can be potentially useful in various fields of medicine and industry, and in the future, they may become a practical and cost-effective competitor against commercial biomaterials currently available on the market.

## 1. Introduction

Microbiologically synthesized cellulose (bacterial cellulose, BNC) is a natural and versatile biopolymer of biologic origin, displaying a high-purity and an extraordinary biocompatibility that can be used either alone or in combination with other ingredients, such as polymers and nanoparticles, to vary its structure and function [1]. Among other naturally derived materials, it is distinguished by its very good mechanical properties, such as high elasticity, tensile strength, excellent ability to retain water, and remarkable biodegradability, which give a number of perspectives for further BNC application in numerous medical and industrial areas [2,3]. Moreover, due to the presence of three free hydroxyl groups located in the C2, C3 and C6 positions of cellobiose, this biological material relatively easily undergoes various chemical reactions and structural changes, e.g., etherification, esterification and acylation [3]. The undeniable advantage of bacterial nanocellulose, which itself possesses different commercially attractive features, is its good moldability and the possibility for targeted modification, which opens up new opportunities for BNC utilization [3]. Ultimately, all properties of this biopolymer strongly depend on several factors, including the pH and chemical composition of the growth medium, cultivation time, and operational conditions, as well as selection of an efficiently growing bacterial strain [4]. The relatively huge diversity of cellulose producing microorganisms, as well as the wide variety of cultivation methods (culture media and their supplementation), create a great opportunity to modify and adjust the properties of this bioinspired material and to find new applications for its usage, such as in the medical sector (wound dressings, implants, scaffolds, drug delivery devices), the field of electronics (sensors, energy storage devices, speakers, acoustic membranes, OLEDs), the environmental sector (water treatment, filtration and adsorption techniques), paper engineering, and the food industry (artificial food, additives, food packaging) [3,5,6,7,8,9,10].

Cellulose producing bacteria have been isolated from various environments (fruits, flowers, fermented foods, rotting fruits, temple wash waters, and vinegar) that highly affect the optimal cultivation conditions, such as pH, temperature, and carbon source [11]. *Komagataeibacter* strains (formerly *Gluconacetobacter* and *Acetobacter*), have been already recognized to be one of the most efficient BNC synthetizing microorganisms. The main source of carbon for most strains is glucose (e.g., *K. xylinus* E26, *K. hansenii* 53582, and *K. hansenii* 23770), but some strains can efficiently grow also on such substrates as fructose or sucrose (e.g., *K. hansenii* ATCC 53582), glycerol (e.g., *K. xylinus* E26, *K. hansenii* SI1), lactic acid, and even on various industrial food wastes [12,13,14,15,16]. Ethanol addition to the culture medium in the case of *K. xylinus* E25, *K. xylinus* subsp. *sucrofermentans* BPR3001A, and *K. rhaeticus* K3 was found to increase the efficiency of cellulose synthesis, while in the case of strain *K. hansenii* 53582, the efficiency of biosynthesis was found to be downregulated [15,17,18,19]. It has been shown that for *K. xylinus* E25 that the addition of ethanol is an additional source of energy, while at the same time it reduces the conversion of glucose to the by-product, gluconic acid [15]. Ethanol has been recognized to stimulate the growth of bacterial cells and as a factor causing oxidative stress, which may increase the production of protective exopolysaccharide such as BNC [13].

Depending on the *Komagataeibacter* strain and the culture conditions used during cultivation, the obtained cellulose membranes differ in properties such as thickness; the content of dry cellulose and exopolysaccharides (EPS), which are hardly soluble in water; fiber density; pellicle flexibility; water-binding capacity; the degree of polymerization; and the crystallinity index [8,20,21,22]. Therefore, it is important to properly select the producing strain depending on the intended use in various fields of industry and medicine. Until now, few scientific articles have dealt with the topic of comparing and selecting strains in order to specifically select and prepare the appropriate cellulosic material [8,21,23]. In this work, we present a comparison of selected parameters of cellulose membranes produced by several strains of the *Komagataeibacter* genus in terms of the selection of the most appropriate cellulose producing strain with the greatest potential as a biological material for the food and medical industry. These membranes show properties suitable for the formation of nanocomposites with other biomolecules that significantly changes the properties of chemically inert cellulose and opens new applications for this biomaterial. This approach also gives the possibility of further modification of the obtained composites, enabling additional functionalization in selected directions.

## 2. Results

### 2.1. The Choice of Komagataeibacter Strains Producing Most Adequate Membranes

In order to choose the most adequate *Komagataeibacter* strains for the production of BNC membranes, all the available 13 producing strains, listed in the Table 1, were first assessed for biosynthetic efficiency. At the same time, the membranes obtained during the cultivation were analyzed for basic parameters such as elongation at break and tensile strength. The combination of these factors constituted the first basis for the selection of strains for further analyses.

#### 2.1.1. Biosynthesis Yield

The thirteen cellulose producing strains were cultivated with and without the addition of ethanol, in order to verify the influence of this agent on the BNC production yields. Among all of the cultures, one underwent contamination (the 11th strain, from ŁOCK collection of pure cultures, see Table 1) and no cellulose was obtained. This strain, therefore, was excluded from further experiments. The rest of the results are presented in Figure 1. When analyzing the obtained outcomes, it is clearly noticeable that the addition of ethanol has a significant effect on production yields, either negatively or positively, depending on the strain. The adverse effect can be observed for the 1st, 4th, and 7th strain, while the growth of the obtained cellulose masses was found in the rest of cultures, with the highest impact for the 3rd, 6th, 8th, and 9th strain. In conclusion, the greatest yields were obtained for the 1st and 4th strains cultivated without ethanol as well as for the 6th, 8th, and 9th strains when cultured in the presence of ethanol. For the 3rd, 5th, 12th, and 13th microorganisms, the results were slightly lower when supplemented with alcohol.

#### 2.1.2. Mechanical Properties

This second stage of bacteria selection was aimed at choosing BNC-synthesizing strains characterized by the best mechanical strength parameters. Young’s modulus, maximum load, and tensile strength were chosen as the most important mechanical parameters taken into account during the analysis. Two variants of cultivation, with the addition of ethanol (Et annotation in the culture name, Figure 2b) and without supplementation (Figure 2a) were taken into account. Based on the results summarized in Figure 2, it can be concluded that the biomaterials with the highest strain at break were that produced by the cultures numbered 1, 4 (Figure 2a), 3Et, 5Et, 6Et, 8Et, 9Et, and 12Et(Figure 2b). The corresponding Young’s modulus values were 5.71 GPa, 4.81 GPa, 6.99 GPa, 4.88 GPa, 7.86 GPa, 8.16 GPa, 6.82 GPa, and 7.69 GPa, respectively. The strains producing cellulose with the highest tensile strength were 1 (121 MPa), 4 (102 MPa), 6Et (145 MPa), 8Et (128 MPa), and 12Et (123 MPa). On the other hand, producers that synthesize membranes of bacterial cellulose, the breaking of which required the highest maximum force, were 1, 4, and 6Et. Their corresponding maximum load values were 250 N, 245 N, and 217 N. Basing on the conducted analyses of cellulose production yield and mechanical properties the following six strains and cultivation variants were selected for further cultivations: 1, 4, 6Et, 8Et, 9Et, and 12Et.

### 2.2. The Properties of BNC Synthesized by Chosen Strains

#### 2.2.1. Physical and Physico-Chemical Parameters

##### Dry Mass Content

Bacterial cellulose membranes produced by six specially selected *Komagataeibacter* strains in seven-days-long stationary cultures were subjected to a detailed comparative analysis. Characterization of the obtained biological materials began with the determination of dry mass content, and the results are presented in Figure 3a. This study showed that the membranes synthesized by the 4th and 1st strains had the highest dry mass content (3.1% and 2.5%, respectively). Such results indicated that these biomaterials, although much thinner, possessed a denser fibril packing and a greater amount of cellulose, compared to other tested samples. In contrast, the more gel-like bacterial cellulose derived from the cultures numbered 6Et, 8Et and 9Et contained less dry mass, indicating a higher degree of hydration (Figure 3b). These results and observations are consistent with the SEM images previously published by Jacek et al. [24].

##### Water Retention and Adsorption Capacity

The ability of the bacterial cellulose membranes synthesized by chosen *Komagataeibacter* strains to retain water were determined using a modified centrifugation procedure (see Section 4.4.1). In our study, this property was expressed as a percentage of the sample weight after liquid separation. As shown in Figure 4a, the 1st and 4th strains synthesized BNC films with a significantly higher WRC in comparison to other analyzed materials. Their WRC values were 19.3% and 18.4%, respectively. On the other hand, the lowest WRC was obtained for the BNC membranes produced by the strain isolated from balsamic vinegar (12Et, 7.3%). These variances in water retention capacity can be explained by the different packing densities of these biofilms [25,26]. It was reported by Ul-Islam et al. that less porous materials can retain liquid longer in the cellulose matrix [27]. The compact structure binds water molecules more efficiently due to the stronger hydrogen bonding interactions. Moreover, the tight fiber arrangement results in a smaller amount of free (unbonded) water [25,28]. Therefore, in this case, the two times lower WRC values are highly connected with looser structure and lower cellulose content in the films. The opposite trend was observed for another water related parameter, namely water adsorption capacity (WAC). In this study, the bacterial cellulose membranes obtained from stationary cultivation were submerged in distilled water to determine their swelling properties. From analyzing the results, presented in Figure 4b, it is noticeable that the materials produced by the cultures numbered 6Et, 8Et, and 9Et had the highest adsorption capacity. The WAC of these BNC films were quite similar and reached values of 46.8 ± 1.0%. In contrast, the compact materials produced by microorganisms numbered 1 (31.5%) and 4 (26.8%) exhibited lower adsorption properties. Such results might be explained by the differences in fiber morphology and structure between cellulose films synthesized by these *Komagataeibacter* strains [24]. The quantity of water molecules penetrating the cellulose matrix and diffusing into amorphous regions depends mainly on the strength of the physical adsorption and on the number and size of pores in BNC-based materials [29]. Therefore, bacterial cellulose membranes with looser structure have more water trapping sites, which correspond to their better swelling properties.

##### Dye Adsorption

The dye adsorption properties of bacterial cellulose were investigated using a compound belonging to the group of anionic azo dyes, Congo Red (CR). This synthetic substance is commonly applied in multiple experiments due to its structural specificity and high affinity for polysaccharides [30]. It is known that CR molecules interact with cellulose fibers through non-covalent interactions, including hydrogen bonding, van der Waals forces, and electrostatic and hydrophobic interactions [31,32]. In this investigation, the amount of dye loaded (adsorbed) after soaking was compared with the concentration remaining in BNC after the rinsing process, thus assessing the binding strength. Additionally, based on the obtained results, the Congo Red adsorption capacity (CRC) of bacterial cellulose was determined. As shown in Figure 5a, the 1st and 4th strains synthesized BNC films that adsorbed the highest amount of dye from aqueous solution, 4.9 mg/L and 5.9 mg/L respectively. Moreover, after a 24 h desorption process, the quantity of Congo Red retained by these solid membranes was greater than 55%, indicating a stronger bonding to cellulose fibers compared to other analyzed materials. Such observations were also confirmed by the calculated CR adsorption capacities (Figure 5b). The results revealed that the more gelatinous bacterial cellulose derived from the cultures numbered 6Et, 8Et, and 9Et exhibited weaker dye adsorption properties. Their CRC values were quite similar and reached 1.96 ± 0.02 mg/g. The lowest Congo Red adsorption capacity was obtained for the bacterial nanocellulose produced by the strain isolated from balsamic vinegar (12Et, 1.7 mg/g). This biomaterial bound only 40% of the dye concentration initially taken from the aqueous solution (Figure 5a). The CR adsorption-desorption studies suggested that the greater dye adsorption capacities can be attributed to a tighter fibril arrangement of bacterial cellulose. The compact BNC structure allows for better retention of Congo Red molecules in the cellulose matrix compared to biomaterials with a more relaxed architecture. Such observations are directly correlated with previous results.

#### 2.2.2. Mechanical Properties

Comparative analysis of bacterial nanocellulose membranes synthesized by the chosen strains was supplemented with tensile and compressive strength measurements. The assessment of the material’s susceptibility to plastic deformation determines its usefulness in various fields, including regenerative medicine, tissue engineering, textile, packaging, and paper industries. In this investigation, tension and compression tests were conducted to evaluate the intrinsic mechanical characteristics of the examined membranes, and at the same time, to confirm the previous observations that the cellulose derived from the 1st and 4th strains had a denser, more compact and, consequently, were less susceptible to plastic deformation structure. Herein, the measurements were performed on BNC films synthesized in 100 mL Erlenmeyer flasks filled with 80 mL of liquid SH medium. This bioreactor change resulted in obtaining thicker materials than those utilized in the strain screening stage. The outcomes of the measurements are presented in Figure 6.

Analyzing the results reveals that the bacterial cellulose membranes synthesized by the 1st and 4th strains were less prone to stress-induced deformation as compared to other tested materials. The maximum force required to result in their breakage was 213 N, which corresponded to an applied load approximately 35% greater than those used for BNC produced by the cultures numbered 6Et, 8Et, and 9Et. In contrast, the obtained F_max_ values for compression tests did not reveal an analogous correlation and maintained at almost the same level of 1040 ± 5 N. Moreover, the results demonstrated that the denser cellulose films derived from the 1st and 4th strains had significantly a higher Young’s modulus, which directly indicates the material’s resistance to plastic deformation. The values for their breaking strength measurements were 3.1 GPa and 3.8 GPa for the 1st and 4th strains, respectively. In other words, the achieved Young’s moduli suggested a greater rigidity of these biological materials, and thus a better resistance to stretching. In addition, it can be observed that BNC films synthesized by the strain isolated from balsamic vinegar (12Et) were also characterized by relatively high values of the elasticity coefficient for tension tests (2.91 GPa). The stress–strain curves showed that bacterial nanocellulose membranes with looser structures (6Et, 8Et and 9Et) exhibited more inelastic behavior (Young’s modulus ~1.13  ± 0.15 GPa). Their reported relative elongation at breakpoint exceeded a value of 10%. The Young’s modulus observed under compressive load displayed the same tendency. As shown in Figure 6b, among the examined bacteria, *K. hansenii* ATCC 53582 (1) and *K. xylinus* E26 (4) produced BNC films that were highly resistant to compressive deformation. For instance, the elasticity coefficient reported for E26-derived cellulose was approximately 3.2 MPa. At the same time, the more gelatinous materials synthesized by the cultures numbered 6Et, 8Et, and 9Et (*K. hansenii* H3, *K. rhaeticus* K4 and *K. rhaeticus* K3) exhibited values of the Young’s modulus an order of magnitude lower under compressive load (320 ± 80 kPa), as compared to other analyzed samples. The mechanical performance studies revealed that BNC membranes produced by the 1st, 4th, and balsamic vinegar isolated (12Et) strains were characterized by the highest breaking strength (113 MPa, 115 MPa and 108 MPa, respectively). This outcome indicated a greater ability to withstand applied tensile stress without damaging the sample. Moreover, 12Et-derived cellulose membranes also exhibited the highest compressive strength, which reached a value of 806 ± 34 kPa. The conducted research revealed a strong positive correlation between the cellulose content and the mechanical properties of BNC. The denser and more compact fibril arrangement of bacterial nanocellulose membranes provides better strength characteristics, while the looser biomaterials with high water content were more susceptible to plastic deformation.

### 2.3. Composites of BNC with Selected Additives

In order to choose the most appropriate *Komagataeibacter* strains for the fabrication of bio-inspired composites, six previously selected microorganisms and their cultivation variants (see Section 2.1.2) were assessed for the physical properties of their membranes. During this third screening step, the following characteristics of the cellulosic biomaterial were taken into account: film compactness, structural integrity, appearance, and thickness. Finally, three efficiently growing strains with the potential to produce durable and functional nanocomposites were selected, i.e., *K. hansenii* H3 (6Et), *K. rhaeticus* K4 (8Et), and the one isolated from balsamic vinegar (12Et). The BNC membranes synthesized by chosen microorganisms are presented in Figure 7.

Composites based on microbial cellulose belong to a new generation of biological materials, which can be widely used for both industrial and medical purposes. They consist of a three-dimensional BNC matrix acting as a scaffold and reinforcing compounds that give it specific physico-chemical and biological properties. The introduction of specific functional groups to chemically inert cellulose allows for the creation of composites with desired physical, chemical, and mechanical features which determine the application potential of this biological material. Herein, bacterial cellulose was modified at the stage of its biosynthesis (in situ method) by supplementing the growth medium with various concentrations of additives, such as citrus pectin (CP), apple pectin (AP), wheat starch (WS), polyvinyl alcohol (PVA), and polyethylene glycol (PEG). The addition of these polymeric compounds was intended to utilize their specific individual characteristics to improve or impart new functional properties to this biological material. This one-step strategy allows the added compounds to become part of the growing bacterial cellulose fibril network, resulting in stable bio-inspired composites [3]. The obtained BNC-based materials were subjected to a detailed analysis in order to assess the impact of polymeric additives on the physical, physico-chemical, and mechanical performance of the cellulose matrix, as well as the possibilities of their potential application.

#### 2.3.1. Physical and Physico-Chemical Parameters

##### Dry Mass Content

Characterization of the composite materials began with determination of dry mass content by a standard gravimetric method. The obtained physical parameters for bacterial cellulose modified with polymer nanoparticles were analyzed and compared to their native counterparts, as a control. The final outcomes of the measurements are shown in Figure 8A. On the basis of the collected data, it can be concluded that the dry mass content of composite materials increased with the additive concentration. Moreover, in each case, the basic weight of the modified BNC films was greater than the control films. Among the tested materials, BNC/PVA and BNC/PEG composites had the highest dry mass content. Their corresponding values varied within the range of 2.5–3.0%. Also, organoleptic observations supported these results (Figure 8B). Bacterial cellulose membranes with polyvinyl alcohol and polyethylene glycol had a compact, hard, and homogeneous structure with a visible layer of reinforcing compound. In contrast, the more gel-like, soft and clearly delicate BNC materials produced in the wheat starch-modified medium (BNC/WS) had the lowest dry mass content.

##### Water Retention Capacity (WRC)

Subsequently, water retention and adsorption analyses were carried out in order to determine the effect of various polymeric compounds on the physico-chemical properties of the obtained BNC composites. All tested biomaterials showed the same low WRC values for the smallest percentages of reinforcing additives. The results followed a tendency to intensify the water retention capacity of BNC composites with increasing concentration of the supplemented compounds, and thus the dry mass content. For instance, the bacterial cellulose membranes produced by *K. rhaeticus* K4 (8Et) reinforced with citrus pectin reached WRC values of 11.3%, 12.5%, and 14.5% for 0.1%, 0.5%, and 1.0% of this plant heteropolysaccharide, respectively. In addition, the water retention capacity of the modified BNC films was mostly higher than the native ones (Figure 9). Based on the collected data, it can be concluded that BNC/PEG had the greatest ability to retain water, which is consistent with the hydrophilic nature of the added compound. The presence of polyethylene glycol in the culture medium provided a statistically significant impact on the tested physico-chemical parameters of the composite. For instance, the WRC obtained for these biomaterials reached 14.6 ± 0.5% at the highest PEG concentration (5.0%).

##### Water Adsorption Capacity (WAC)

Another important feature of bacterial cellulose composites from an industrial and medical application perspective was their water adsorption properties. For instance, maintaining proper moisture content inside the packages and removing exudates from foodstuff are key aspects that must be fulfilled to prevent microbial contamination [3]. The results revealed that BNC/PEG composites were characterized by the highest water adsorption capacity with respect to the other tested materials. The maximum WAC attained for bacterial cellulose films synthesized by *K. rhaeticus* K4 (8Et) and modified in situ with 5% polyethylene glycol was 66.7 ± 0.7% (see Figure 10). In the case of BNC/PEG samples, their water sorption efficiency increased with the concentration of the reinforcing compound. Moreover, it was found that even addition of a low content of polyethylene glycol to the growth medium improved the WAC of the obtained BNC-based composite materials as compared to unmodified (native) bacterial cellulose films. On the other hand, the biomaterial exhibiting the lowest water adsorption capacity was BNC/PVA. For instance, the nanocellulose composites produced by *K. hansenii* H3 (6Et) reached the following WAC values for successively increasing concentrations of polyvinyl alcohol: 18.5%, 24.3% and 32.6%. In addition, it can be noticed that the presence of PVA caused a statistically significant decrease in the water adsorption properties of the modified biomaterials, when compared with native BNC membranes. Similarly, in the case of other analyzed composites (BNC/CP, BNC/AP, BNC/WS), supplementation of the standard culture medium with these polymeric compounds mostly resulted in a reduction of the WAC values.

##### Dye Adsorption Capacity (CRC)

Congo Red loading experiments were performed to evaluate and compare the ability of the analyzed BNC-based nanocomposites to bind hazardous dye molecules from aqueous solutions. This study allowed to verify the suitability of these green nanomaterials as dye adsorption/removal pads or delivery carriers for bioactive agents that could be further applied in the packaging, biotechnology, and medical industries, among others. As shown in Figure 11, supplementation of the culture medium with polymeric compounds resulted in a tremendous improvement in the CR dye uptake properties of cellulosic composite materials. In each case, the modified BNC membranes had a greater Congo Red adsorption capacity than the controls. For instance, 8Et-derived bacterial cellulose reinforced with 1% (*w*/*v*) apple pectin reached a CRC value of 14.4 mg/g, which was more than seven times higher than its native counterpart. This investigation demonstrated that BNC/CP and BNC/AP membranes were able to adsorb significant quantities of Congo Red particles from aqueous solution during a one hour contact time, which makes them the most effective dye-removing composite materials among those analyzed (see Figure 11). Moreover, the results followed a tendency to intensify the CR adsorption performance with increasing pectin concentration.

##### Water Vapor Permeability

The barrier properties of bio-based packaging materials, such as the water vapor transmission rate (WVTR), play a crucial role in maintaining quality, microbiological safety and proper shelf life of food products [3]. Controlling moisture permeability is important to keep foodstuff freshness and prevent them from mold and mildew deterioration. WVTR is a standard parameter that determines a material’s ability to pass water vapor through its surface under certain circumstances [33]. Since a reduction in moisture permeability is particularly desirable for the potential application of bacterial cellulose films in the food packaging industry, the effect of the incorporation of polymeric compounds into BNC matrices was investigated. Figure 12 shows the WVTR values of native bacterial nanocellulose membranes as well as BNC/CP, BNC/PEG, and BNC/PVA composites. The addition of citrus pectin and polyvinyl alcohol to the culture medium reduced the water vapor transmission rates of the obtained composites by 35% and 16%, respectively, in comparison to their native counterparts. It is known that WVTR depends on the thickness, composition, available surface area, homogeneity, and permeability of the materials [34]. Therefore, this statistically significant decrease in moisture vapor transmission rate may be explained by the altered porosity and higher density of the composites as a result of loading their 3D cellulose network with reinforcing agents. Overall, the compact structure of the biomaterials slows down the water vapor exchange between foodstuffs and the surrounding environment, thus lowering the WVTR values [35,36]. On the other hand, BNC/PEG membranes exhibited the highest moisture vapor permeability among the analyzed membranes (19.6 ± 0.4 g/m^2^∙h). Thus, the barrier properties of these composites were reduced by 4% as compared to unmodified bacterial nanocellulose films. Such outcomes may be related to the hydrophilic nature of polyethylene glycol molecules. The greater number of hydroxyl groups present in BNC/PEG films allowed more water molecules to enter the bacterial cellulose structure [37]. Moreover, the incorporation of plasticizers, such as polyethylene glycol, led to the reduction of hydrogen bonds between BNC fibers, effectively increasing the porosity of this biomaterial [37].

##### Fourier Transform Infrared Spectroscopy (ATR-FTIR)

The chemical composition of native and modified bacterial cellulose samples, including their specific structural and conformational differences, was determined by attenuated total reflection Fourier transform infrared (ATR-FTIR) spectroscopy. This analytical method was used to gather information about the functional groups and chemical bonds present in the tested materials, and consequently to confirm the effectiveness of BNC functionalization. The collected IR spectra are shown in Figure 13. It can be clearly seen that the absorption bands characteristic for cellulose, occurring in two wavenumber ranges of 3660–2800 cm^−1^ and 1800–650 cm^−1^, are visible in all analyzed materials. Nevertheless, the modification of bacterial cellulose with various polymeric compounds affected the position, width, shape, and intensity of these typical IR signals, as well as led to the appearance of new peaks corresponding to the added functional groups.

The ATR-FTIR spectra of BNC membranes contained a strong absorption band in the range of 3500-3200 cm^−1^, which was attributed to O–H stretching vibration of cellulose I [13]. The narrow peak at 3346 cm^−1^ represents the strength of the intramolecular 3(O)H⋯O(5) hydrogen bond [38]. On the other hand, the absorption band at 2896 cm^−1^ is related to C–H asymmetric stretching of aliphatic CH_2_ and CH_3_ groups [39]. Another relatively intense peak is located at 1650 cm^−1^ and corresponds to the bending vibration of water molecules (H–O–H) absorbed in the biopolymer structure [40]. Two characteristic signals at 1428 cm^−1^ and 1315 cm^−1^ are attributed to CH_2_ symmetric stretching and out-of-plane wagging of the CH_2_ groups, respectively [41]. According to the literature data, the absorption band at 1428 cm^−1^ is attributed to the highly ordered (crystalline) regions in bacterial nanocellulose [38]. The ATR-FTIR spectra also shows four intense IR signals in the fingerprint region (1160–1033 cm^−1^), which correspond to C–O–C antisymmetric bridge stretching of β-1,4-glycosidic bond, C–C, and C–O stretching vibrations, C–O–H bending of carbohydrates, and C–O–C pyranose ring skeletal vibration [41]. The narrow absorption peak at 899 cm^−1^, known as the “amorphous band” is related to the out-of-phase ring stretching of β-1,4-glycosidic linkages in bacterial cellulose [41].

Compared to native BNC, the ATR-FTIR spectra of the analyzed BNC composites revealed some changes in signal intensity, peak shifts, and the appearance of new absorption bands typical for the added reinforcing compounds. In the case of pectin-modified bacterial cellulose membranes, the peak at 3346 cm^−1^, attributed to O–H stretching vibrations in polysaccharides, became much broader. This width difference can be related to the interlinkage of pectin and BNC fibers through H bonding, which is in agreement with the literature data [42]. As shown in Figure 13, one of the most intense IR signals for BNC/CP and BNC/AP films were the carboxylate antisymmetric stretching bands (COO^–^) found in the range of 1630–1600 cm^−1^ [43]. Moreover, differences in the ATR–FTIR spectra of these composites compared to those registered for unmodified material appeared also in the region of lower wavenumbers (1330–830 cm^−1^), which was dominated by the stretching vibrations of C–O, C–C, and ring structures characteristic of pectin [43]. Another peak, not present in the native BNC spectrum, was observed at 957 cm^−1^. This band was associated with C–O bending deformation [43]. The described changes were caused by the presence of hydroxyl, carboxyl, methoxy, and acetyl groups in pectin molecules. The spectra of PVA-modified bacterial cellulose membranes were quite similar to their native counterparts. The main differences that could be observed occurred in the hydroxyl and carbonyl regions. For instance, the intensive band in the range of 3500–3200 cm^−1^ was broadened and slightly shifted towards lower wavenumbers, which proved that the formation of intermolecular hydrogen bonds between the hydroxyl groups of bacterial cellulose and polyvinyl alcohol occurred. The success of BNC modification was also confirmed by the appearance of a peak at around 1700 cm^−1^, assigned to carbonyl groups (C=O) present in the partially hydrolyzed acetate residues in PVA [44]. Moreover, another absorption band characteristic for polyvinyl alcohol, absent in the native BNC spectrum, was observed at 836 cm^−1^. This IR signal was associated with CH_2_ deformation vibrations of vinyl polymers [45]. Similarly, in the case of BNC/PEG composites, the absorption band in the range of 3500–3200 cm^−1^ and 1650 cm^−1^ were significantly increased. This happened due to large amounts of hydroxyl groups in polyethylene glycol molecules [37]. The peak at 2895 cm^−1^, characteristic of the aliphatic C–H stretching vibration, was also intensified. Likewise, the IR signal intensity in the region of lower wavenumbers was greatly enhanced. These outcomes were consistent with the data reported in the literature [46,47]. All the described differences between the native and modified BNC spectra indicated intermolecular interactions occurred through the hydroxyl groups of bacterial cellulose and PEG. Thus, the ATR-FTIR analysis confirmed the presence of functional groups characteristic for the added compounds and their successful incorporation into the BNC fibrillar nanostructure.

#### 2.3.2. Mechanical Properties

The ultrafine nanofibrillar network and well-ordered structure of bacterial cellulose ensure its relatively good mechanical properties. Moreover, it is possible to improve or modify the tensile and compressive strength of this biomaterial by supplementing the growth medium with reinforcing additives and alternative energy sources, such as organic acids, vitamins, and alcohols [3,13]. Hence, the influence of citrus and apple pectins, wheat starch, polyvinyl alcohol, and polyethylene glycol on mechanical parameters of BNC membranes was investigated.

Mechanical characterization of the obtained biocomposites began with the examination of their stretching behavior. All the collected data from breaking strength measurements are presented in the Appendix A. The outcomes provided clear evidence that the analyzed in situ modifications of bacterial cellulose had a statistically significant impact on the mechanical properties of these polymeric biomaterials. The maximum force (F_max_) that BNC films can withstand during stretching is one of the most important parameters examined, clearly displaying the mechanical strength differences between the analyzed samples. Based on the results summarized in Figure 14, it can be concluded that BNC/PEG and BNC/PVA were less susceptible to stress-induced deformation when compared to the other tested biomaterials. Specifically, the maximum force required to induce their breakage varied within the range of 250–320 N at the highest additive concentration (5.0%). Moreover, in each case, bacterial cellulose composites with polyvinyl alcohol and polyethylene glycol were characterized by higher resistance to stretching compared to the control (unmodified) ones, as clearly shown in the graph below. For instance, the maximum load required to break 8Et-derived BNC reinforced with 3% PVA was over two times higher than that utilized for their native counterparts.

The obtained composites based on bacterial cellulose were also subjected to compression testing. Young’s modulus, the maximum crushing force, and compressive strength were the most important mechanical parameters considered in this analysis. A complete summary of the data collected from the compression measurements is provided in Appendix A. Analysis of the results reveals that the modifications conducted in this experimental work led to changes in the mechanical characteristics of BNC membranes. All tested composites displayed a statistically significant reduction in the maximum load and, at the same time, an increase in Young’s modulus in comparison to their native counterparts. The compression strength that the modified bacterial cellulose films can bear before fracturing is a representative parameter that well illustrates the differences between the analyzed biomaterials. As shown in Figure 15, BNC/PVA and BNC/PEG were the most resistant to compressive deformation. Their corresponding strength (C_S_) values were 818 ± 54 N and 748 ± 55 N, respectively, at the highest additive concentration (5.0%). On the other hand, BNC films synthesized in the culture medium modified with wheat starch were the most susceptible to plastic deformation caused by compression forces. The evidence explicitly supporting this observation was 8Et-derived bacterial nanocellulose reinforced with 0.1% WS, which achieved a compression strength value almost 24 times lower than its native counterpart. Such results may be related to the fact that the addition of gelatinized starch changes the viscosity of SH medium, which in turn reduces the motility of bacterial strains and negatively affects binding of cellulose molecules during BNC synthesis [48]. Consequently, the formed nanofibrillar network showed the weakest mechanical behavior. Moreover, from among all the analyzed variants, the strain isolated from balsamic vinegar (12Et) produced BNC composites with the best mechanical properties tested under compressive load.

#### 2.3.3. Bacteriostatic Properties

In this study, the agar disc diffusion method was used to investigate the potential bacteriostatic properties of the chosen BNC composites against two representatives of Gram-negative (*Escherichia coli*) and Gram-positive (*Bacillus subtilis*) bacteria. The results of antibacterial activity of the analyzed biomaterials after 24 h of exposure are presented in the Appendix A. It is clearly visible that no zones of microbial growth inhibition appeared around the samples discs. This indicates that the examined BNC composites did not show bacteriostatic properties against these two microorganisms.

## 3. Discussion

In the literature, one may find numerous outcomes characterizing various strains of cellulose producing bacteria. Each strain delivers cellulose films at variable yield and with different structural parameters [8]. While the efficiency has a direct impact on production costs, the properties of BNC membranes from different strains have a determining significance for their potential application and effectiveness. The *Komagataeibacter* strains investigated in this study synthesized bacterial cellulose with yields ranging from 0.56 g/L to 6.51 g/L. The obtained efficiencies are comparable to those reported for other cellulose-producing bacteria using glucose as a carbon source in the standard SH medium at 30 °C [14,20,49,50]. Such comparisons, however, are difficult to be performed and are often less reliable, since each study’s cultivation of microorganisms and analytical methods are carried out following differing procedures. Therefore, In the investigation provided by us we unify the procedures and analyze several BNC-producing strains at once so as to compare the resultant membranes. Noticeable in most of the analyzed cases was that the yield of cellulose biosynthesis was significantly influenced by the supplementation of ethanol, an enhancer promoting BNC biosynthesis and bacterial growth [15]. In this study, the addition of this agent to the culture medium increased the yield 2-3 fold for *K. hansenii* strains up to approximately 5 times the yield without the supplementation for *K. xylinus* E25. Still, the greatest yield was obtained for *K. xylinus* E26 (4) without ethanol addition, which may suggest that this strain is very promising not only from a financial point of view, but also when looking at the membrane properties of this strain, which can be observed in further examinations.

The selection of suitable strains for the production of both bacterial cellulose membranes and their composites is very important step affecting the final quality, morphology, properties of the obtained biomaterials and, consequently, their further application. In this study, based on the BNC biosynthetic efficiency, mechanical strength and physical features (i.e., compactness, structural integrity, appearance and thickness) of the synthesized films, three *Komagataeibacter* strains and their cultivation variants (*K. hansenii* H3 (6Et), *K. rhaeticus* K4 (8Et), and the one isolated from balsamic vinegar (12Et)), were chosen for bio-inspired nanocomposites fabrication. The organoleptic observations of the selected BNC-producing microorganisms revealed that the cultures numbered 6Et and 8Et synthesized cellulose membranes with a looser and more gelatinous structure than the 12Et-derived ones (Figure 7). This relaxed form, resulting in an increased pore volume and surface area, had a direct impact on the physico-chemical and mechanical performance of these biomaterials. The 6Et and 8Et variants produced BNC films that were characterized by a significantly higher water adsorption capacity while, at the same time, having a lower tensile and compressive strength than the other examined membranes. These findings correlate well with the previously published research on bacterial cellulose with loose fibril arrangements [14,25,51,52]. The higher porosity of BNC-based materials allows them to be more efficiently loaded with various reinforcing and bioactive compounds. Moreover, it may also facilitate the process of their release [6]. Such outcomes suggest that bacterial cellulose membranes synthesized by *K. hansenii* H3 (6Et) and *K. rhaeticus* K4 (8Et) strains are potentially good candidates for a wide range of applications, including wound dressings, carriers for controlled drug delivery, scaffolds for tissue engineering, active and intelligent food packagings, absorbing materials, environmental remediators, and high-performance electronic devices [3,53,54,55,56,57]. On the other hand, the strain isolated from balsamic vinegar (12Et) produced BNC films characterized by a more compact and homogenous structure compared to those analyzed above. Therefore, they exhibited better mechanical properties under breaking and compressive loads (Figure 6). In addition, it can be concluded that the 12Et variant synthesized bacterial cellulose nanocomposites with better efficiency, as evidenced by the differences in the dry matter content (Figure 8A). These properties suggest that BNC films produced by *Komagataeibacter* sp. isolated from balsamic vinegar (12Et) could be applied successfully as scaffolds for cells cultivation, dental and medical implants, artificial skin, blood vessels, bone and cartilage substitutes, and highly resistant packaging, papers, and textiles [7,54,58,59,60].

Bacterial cellulose synthesized by various *Komagataeibacter* strains is characterized by different structural and morphological features, and thus, specific physico–chemical and mechanical properties. Taking into account the individual characteristics and limitations of BNC membranes, they can be concretely modified to obtain biocomposites for the desired applications. Combination of the natural attributes of this nanofibrillar polymer matrix as well as the physico-chemical, mechanical, and biological properties of reinforcing compounds allows for the improvement of its parameters or to create materials with novel useful features. In this study, bacterial cellulose was modified by supplementing the basal medium with various concentrations of citrus pectin (CP), apple pectin (AP), wheat starch (WS), polyvinyl alcohol (PVA), and polyethylene glycol (PEG). The ATR-FTIR analysis confirmed the successful incorporation of these polymeric compounds into the three-dimensional nanocellulose network. The collected spectra indicated the presence of hydroxyl, carbonyl, carboxyl, methoxy, and acetyl groups that modified the original properties of chemically inert BNC (Figure 13). In general, the addition of pectin during biosynthesis, regardless of its origin, resulted in a decrease in the tensile and compressive strength of the obtained BNC-based composites. Such outcomes are consistent with those published by Augustin et al. [42] and Szymańska-Chargot et al. [61]. Moreover, as shown in Figure 9, BNC/CP had a higher water retention capacity when compared to other analyzed materials. It is known that the presence of hydrophilic substances promotes significant changes in the WRC values [6,25]. Citrus pectin is a complex heteropolysaccharide extracted from dried fruit peel which, due to its high hydrophilicity, is commonly used as a gelling, thickening, and stabilizing agent in the food sector [42,62]. Congo Red uptake experiments demonstrated that pectin-modified bacterial cellulose films exhibited statistically significant higher dye adsorption abilities than the control ones. The CRC values increased with growing concentration of such thickening compounds. Such outcomes can be attributed to the strong synergistic effect of hydrogen bonding and electrostatic interactions occurring between the amine-containing dye and pectin-modified cellulose fibers. There are various studies related to surface functionalization which show that the incorporation of hydrophilic functional groups, such as -OH and -COOH in the polysaccharide structure enhances their adsorption/removal efficiency for organic dyes such as Congo Red [63,64]. These properties suggest that BNC/CP and BNC/AP composites can potentially be used as matrices for drug release systems or enzyme immobilization, effective dye-removing materials, and adsorption/removal pads in the packaging industry. The addition of polyvinyl alcohol to SH medium caused a statistically significant decrease in the water adsorption capacity of the obtained biomaterials, however, their mechanical characteristics were improved. These findings are strongly connected with the structural features (e.g., film’s compactness and toughness) and, consequently, the high dry mass content of the tested composites. This effect of PVA supplementation on the physico-chemical and mechanical properties of the obtained BNC membranes was also recently proven by Długa et al. [44]. The BNC/PVA prepared according to the in situ method may be useful in biomedical, biotechnological and packaging applications that demand materials with improved strength parameters, such as wound dressings, tissue engineering, regenerative medicine, e.g., scaffolds for cells cultivation, and medical implants, and durable packaging. A similar effect of the addition of a polymeric compound on mechanical properties was observed for BNC/PEG composites. In general, they were less susceptible to stress-induced deformation than the other examined biomaterials. The positive impact of polyvinyl alcohol and polyethylene glycol on the mechanical behavior of bacterial cellulose can be attributed to their plasticizing properties. Some authors have reported that such polymeric additives not only cover, but also penetrate the polysaccharide structure, forming hydrogen bonds with cellulose fibers [47]. Consequently, they improve the flexibility and mechanical resistance to stretching of BNC membranes, which may be potentially useful in tissue engineering, regenerative medicine, and packaging applications [44,65]. Moreover, the presence of polyethylene glycol in the basal SH medium positively influenced the water adsorption/retention capacities of the synthesized bacterial cellulose films. The obtained results indicated that the water retention and adsorption capacity of the analyzed composites based on microbial cellulose strongly depended on both the structural morphology of BNC matrix (e.g., pore size, distribution, available surface area) and the nature of the added reinforcing compounds. The presence of hydrophilic substances promotes significant changes to the WRC and WAC values [6,25]. Polyethylene glycol as a humectant (moisturizing ingredient) found in many cosmetic products is characterized by strong hygroscopic properties; therefore, BNC/PEG composites had a noticeably higher water retention and adsorption capacity compared to other tested materials. These observations are also supported by various studies [37,66,67]. This type of plasticizer creates a layer that covers bacterial nanocellulose fibrils, and by reducing the hydrogen bonds between them, it leads to the formation of a highly porous biomaterial that is able to hold more water molecules [37]. Other studies have also confirmed improved physio-chemical and mechanical properties of BNC/PEG films [37,47,68]. Due to these properties, polyethylene glycol-modified bacterial cellulose membranes are potential candidates for a wide range of applications, including wound dressing, moisturizing masks, highly-absorbent materials, as well as durable green packaging and scaffolds for tissue engineering.

Comparative analysis of bacterial cellulose synthesized by various *Komagataeibacter* strains confirmed that their production efficiency and subsequent physico-chemical/biomechanical properties, differed depending on the chosen BNC-synthesizing microorganism; therefore, in order to obtain the materials for the intended applications, several factors should be taken into account, such as the pH and chemical composition of the growth medium, cultivation time, operational conditions, and the selection of an efficiently growing bacterial strain [4]. Well-selected native bacterial nanocellulose itself is a highly competitive biological material with numerous possible application areas, including medicine, biotechnology, cosmetics, textile, packaging, and paper industries [3,69]. Furthermore, taking into account the natural attributes and limitations (e.g., hydrophilic character and the lack of antimicrobial and antioxidant activities) of bacterial cellulose, it can be specifically modified to improve or provide new useful features. In the literature, one can find many ways to enhance the physico-chemical and mechanical properties of this biological material, such as the addition of reinforcing compounds, plasticizers, stabilizers, other polymers, inorganic and organic nanoparticles, bioactive agents (e.g., antimicrobials, antioxidants, plant extracts, organic acids etc.), therapeutic additives, and proteins [3,4,70,71]. Among them, one of the most attractive possibilities, offering mild conditions of BNC treatment and lack of toxicity, is the utilization in enzymes. This future-oriented approach, based on green technology, offers an opportunity to improve the functionality of these biological materials and create innovative bio-inspired nanocomposites, without limiting any outlooks for their further application.

## 4. Materials and Methods

### 4.1. Komagataeibacter Strains and Cultivation Methods

Bacterial cellulose synthesis was performed by using 13 strains of *Komagataeibacter* genus belonging/made available to the Institute of Molecular and Industrial Biotechnology (IMIB). All of the investigated microorganisms are listed in the Table 1.

Microorganisms were stored at −70 °C. The strains were cultured in standard SH medium (Schramm and Hestrin medium) composed of glucose (20 g/L), yeast extract (5.0 g/L), peptone (5.0 g/L), MgSO_4_ × 7H_2_O (2.5 g/L), Na_2_HPO_4_ (2.7 g/L), and citric acid (1.15 g/L), which was prepared in distilled water at pH 5.7 and autoclaved before use.

Bacteria were activated from stock by cultivation in SH medium, and then supplemented with 2% agar in petri dishes at 30 °C for 5 days. A single colony was then transferred to 5 mL of liquid SH medium with or without the addition of ethanol (1% *v*/*v*) and incubated at 30 °C for 2 days. Standard cultures, depending on the following analyses, were performed in 100 mL and 500 mL Erlenmeyer flasks filled with either 80 or 100 mL of liquid SH medium, respectively. Each time the medium was inoculated with 5% (*v*/*v*) of 2 days pre-culture and incubated for 7 days at 30 °C. Then BNC membranes were then cleaned of residual medium and bacterial cells by rinsing the membranes with water until white, then placing in 1% NaOH followed by neutralization in 1% acetic acid and water until the pH was neutral. All the cultivations of each analyzed strain, designed for particular experiments, were prepared in triplicates.

### 4.2. Composites Preparation Methods

From the six *Komagataeibacter* strains and their cultivation variants selected on the basis of the mechanical parameters of the obtained bacterial nanocellulose membranes (Section 2.1.2), finally, three microorganisms were chosen for the production of BNC-based composites. At this stage, the selection criteria were the physical properties of the biomaterials, including their structural integrity, appearance, and thickness of the films. The following *Komagataeibacter* strains were chosen for the biosynthesis of BNC composites: 6Et, 8Et, 12Et.

Bacterial cellulose composites were prepared according to the in situ strategy by adding different concentrations of compounds to the culture medium at the beginning of the BNC production process. BNC/CP and BNC/AP were obtained by supplementing 100 mL of SH medium with 0.1 g, 0.5 g, or 1.0 g of citrus/apple pectin and 12.5 mM CaCl_2_. The presence of calcium chloride (more precisely Ca^2+^ ions) was required for the formation of the three-dimensional gel network [75]. In order to obtain BNC/SP composites, 50 mL of liquid basal medium was mixed with 50 mL of gelatinized wheat starch at the defined concentration of either 0.1%, 0.5%, or 1.0%. The process of swelling and partial disruption of cereal polysaccharide granules took place at temperatures above 60 °C. In turn, films based on bacterial nanocellulose and polyvinyl alcohol (DP = 3300) were obtained by adding 10 mL of 1%, 3% or 5% PVA solution to 90 mL of liquid SH medium. An accurately weighed amount of the aforementioned powder was dissolved in hot distilled water under continuous stirring. Similarly, BNC/PEG composites were prepared by supplementing 90 mL of the basal medium with 10 mL of 1%, 3% or 5% polyethylene glycol solution. PEG (Mw = 8000) was easily dissolved in distilled water. Table 2 summarizes all the performed modifications of the culture medium.

The modified SH medium supplemented with 1% (*v*/*v*) ethanol was inoculated with 5% (*v*/*v*) of 2 days pre-culture. Bacterial cellulose composites were grown in stationary conditions at the temperature of 30 °C. After 7 days of incubation, the obtained membranes were thoroughly purified with tap water to remove the remaining medium and bacterial cells and then treated with 1% (*w*/*v*) NaOH and neutralized with 1% (*v*/*v*) acetic acid. Afterwards, the BNC composites were washed in distilled water until the pH was neutral. The purified biomaterials were sterilized by autoclaving at 121 °C and stored at 4 °C until further analysis.

### 4.3. Physical Parameters Evaluation of BNC Membranes and Their Composites

#### 4.3.1. Yield of Biosynthesis Process

The BNC production yield was measured as the dry weight of cellulose recalculated per volume of the medium (g/L).

#### 4.3.2. Dry Mass Content

Bacterial cellulose membranes (native and composites) obtained after seven days of stationary cultivation were weighed in a wet state and then dried to constant mass using a vacuum gel dryer (Model 543, BIO-RAD, Hercules, CA, USA). This process was carried out for 2 h at a temperature of 80 °C. Subsequently, BNC membranes were reweighed on analytical balance and dry mass content was calculated from the following equation:(1)$$\mathrm{Dry}\;\mathrm{mass}\;\mathrm{content}\;\left[\%\right]=\frac{m_{\mathrm{dry}\;\mathrm{BNC}\;\mathrm{membrane}}\left[g\right]}{m_{\mathrm{wet}\;\mathrm{BNC}\;\mathrm{membrane}}\left[g\right]}\times 100\%$$

Dry mass content was expressed as a percentage and represents the mean value of at least three replicates. The experiment was conducted for both native BNC membranes and their composites.

### 4.4. Physico-Chemical Parameter-Based Evaluation of BNC Membranes and Their Composites

#### 4.4.1. Water Retention Capacity (WRC)

Water retention capacity is a parameter that describes a material’s ability to retain water and it is defined as the percentage of sample weight after centrifugation of the liquid to its initial value. The WRC of BNC membranes was determined according to the modified method described by Tomer et al. [76].

Wet bacterial cellulose films (native and composite materials) were cut into pieces with masses in the range of 0.3 to 0.4 g. Next, BNC samples were placed into Eppendorf tubes with specially prepared “gauze baskets” (see Appendix A) and centrifuged for 30 s at 6000 rpm (Microcentrifuge Model 5415R, Eppendorf, Hamburg, Germany). The tests were performed at ambient temperature. After centrifugation, the weight of bacterial cellulose was weighed on analytical balance. Water retention capacity (WRC) was calculated from the following equation:(2)$$\mathrm{WRC}\;\left[\%\right]=\frac{m_{\mathrm{after}\;\mathrm{centrifugation}}\left[g\right]}{m_{\mathrm{before}\;\mathrm{centrifugation}}\left[g\right]}\times 100\%$$

This experiment was conducted in at least six replicates for each type of BNC samples.

#### 4.4.2. Water Adsorption Capacity (WAC)

Determination of adsorption abilities was carried out for bacterial cellulose films that had previously been subjected to compression tests. The aim of this process was to compare the swelling properties of native BNC membranes synthesized by different strains as well as their composites. Cellulose samples squeezed out of the water were weighed and subsequently immersed in distilled water. The experiment was performed under stationary conditions at room temperature. After 20 h of rehydration, the change in the BNC membranes’ weight was determined on analytical balance. Water adsorption capacity was established by the following equation:(3)$$\mathrm{WAC}\;\left[\%\right]=\frac{m_{\mathrm{wet}\;\mathrm{BNC}\;\mathrm{membrane}}\left[g\right]-{\;m}_{\mathrm{dry}\;\mathrm{BNC}\;\mathrm{membrane}}\left[g\right]}{m_{\mathrm{dry}\;\mathrm{BNC}\;\mathrm{membrane}}\left[g\right]}\times 100\%$$

The WRC average values and standard deviations were calculated from at least 3 replicates.

#### 4.4.3. Water Vapor Permeability

The water vapor transmission rate (WVTR) of the composite materials was determined gravimetrically in accordance with the ASTM E96 standard (water method) [33]. Briefly, native and modified BNC membranes synthesized by the strain isolated from balsamic vinegar (12Et) were completely dried at 80 °C in a vacuum gel dryer and subsequently conditioned for 24 h at 25 °C before performing the analysis. Next, round-shaped samples with a diameter of 20 mm were precisely mounted on cylindrical cups containing distilled water and secured with a rubber gasket (see Appendix A). The water level was kept at 20 mm from the BNC membrane, which was necessary to avoid contact with the material. After weighing, the measuring vessel was placed into an oven at 25 °C. The mass change was determined on an analytical balance in time intervals for a period of 2 days. The WVTR was calculated from the following equation:(4)$$\mathrm{WVTR}\;\left[\frac{g}{m^{2}\;h}\right]=\frac{\Delta m\left[g\right]}{\Delta t\;\left[h\right]\times A\;\left[m^{2}\right]}\times 100\%$$
where Δm is the weight of water passed through the membrane, Δt represents the time difference between two weight measurements, and A is the exposed surface area of the BNC membrane. The experiment was conducted in triplicate for each composite and compared to the native BNC material.

#### 4.4.4. Congo Red Adsorption Capacity

Wet bacterial cellulose membranes (native and composites) were cut into fragments weighing 0.3–0.4 g and placed into 5 mL of 0.004% Congo Red (CR) solution (40 mg/L). The concentration of the dye used in these CR loading experiments was selected on the basis of the preliminary tests performed in the concentration range of 20–50 mg/L. The prepared samples were then shaken for 1 h at 100 rpm and 25 °C. Afterwards, the absorbance of the remaining dye solution was measured by UV/Vis spectrophotometer (6300 VIS/6305 UV-VIS, Jenway) at 497 nm. The Congo Red concentration adsorbed by wet bacterial cellulose was calculated using a standard dye calibration curve (*y* = 0.0413*x*) and the following equation:(5)$$\Delta C_{\mathrm{adsorbed}}\;\left[\mathrm{mg}/L\right]={\;C}_{0}-{\;C}_{1}$$
where C_0_ is the initial concentration of CR (C_0_ = $A_{497\;\left(before\;adsorption\right)}/0.0413)$ and C_1_ represents the residual dye concentration after adsorption (C_1_ =$A_{497\;\left(after\;adsorption\right)}/0.0413)$.

In the next part of this investigation, the desorption behavior of bacterial nanocellulose loaded with Congo Red was examined. BNC films (native and composites) after the previously described adsorption experiment were gently placed in 5 mL of distilled water in order to rinse the dye from the samples. The process was carried out for 24 h in a shaker (Infors HT Minitron) at 150 rpm and 25 °C. After this time, the absorbance of the supernatant was measured spectrophotometrically at 497 nm. The amount of dye bound to bacterial nanocellulose membranes (C_bound_) was calculated using a standard dye calibration curve (*y* = 0.0413*x*) from the following equation:(6)$$C_{\mathrm{bound}}\;\left[\mathrm{mg}/L\right]=\Delta C_{\mathrm{adsorbed}}-{\;C}_{2}\;$$
where ΔC_adsorbed_ is the dye concentration adsorbed by BNC membrane after 1 h of soaking and C_2_ refers to the Congo Red concentration desorbed from BNC membrane (C_2_ = $A_{497\;\left(after\;desorption\right)}/0.0413)$.

Ultimately, Congo Red adsorption capacity (CRC) was established by the following equation:(7)$$\mathrm{CRC}\;\left[\mathrm{mg}/g\right]=\frac{C_{\mathrm{bound}}[\mathrm{mg}/L]\times \;V\;\left[L\right]\;}{m_{\mathrm{dry}}\left[g\right]}$$
where C_bound_ represents the amount of dye finally bound to bacterial nanocellulose, V is the volume of Congo Red solution, and m_dry_ represents the weight of dried membrane. The experiment was conducted in three replicates for each BNC sample (native and composites).

#### 4.4.5. Fourier Transform Infrared Spectroscopy (FTIR)

FTIR was used to determine the functional groups and chemical bonds present in the composites, thus assessing the effectiveness of BNC modification. Bacterial cellulose membranes were cut into 2 × 2 cm squares and freeze-dried (0.36 mbar, −30 °C, 24 h) prior to the analysis. The FTIR measurements were performed on a Nicolet 6700 spectrophotometer (Thermo Fisher Scientific, Waltham, MA, USA) equipped with an attenuated total reflectance (ATR) attachment under ambient conditions. The spectra of BNC, BNC/CP, BNC/AP, BNC/WS, BNC/PVA, and BNC/PEG were recorded in transmission mode within the range from 4000 to 650 cm^−1^ and a resolution of 4 cm^−1^. The OMNIC Specta software program (version 9.2.86, Thermo Fisher Scientific, Waltham, MA, USA) was used for data analysis.

### 4.5. Mechanical Properties Testing

Bacterial cellulose membranes (native and composites) were examined for tensile and compressive strength using a universal testing machine (Zwick Roell 1 kN, type Xforce HP, S/N:764916, Germany). The mechanical parameters characteristics for BNC, i.e., Young modulus, maximum force (F_max_), stress (σ), and elongation at break (ε), were determined by the TestXpert II software program (version 3.61, ZwickRoell GmbH & Co.KG, Ulm, Germany, 2015). Stress was equal to the loading force, expressed in Newtons (N), divided by the cross-sectional area, measured as the width × thickness of the sample (m^2^). The strain was calculated as ΔL/L_o_ × 100%, where L_o_ is the initial length and ΔL represents the change in original length after breakage. The Young’s modulus values were determined on the basis of the linear stress/strain relationship. After the mechanical test, the compressive and breaking strength of native BNC and their composites were calculated from the following equation:(8)$$\mathrm{Strength}\;\left[N/m^{2}\right]=\frac{F_{\max}\left[N\right]\;}{\mathrm{cross}\;\sec\mathrm{tion}\;\mathrm{area}\left[m^{2}\right]}$$

The measurements were performed in at least three replicates for each type of mechanical testing, in an air-conditioned laboratory, at a temperature of 20 °C.

#### 4.5.1. Breaking Strength Measurements

Before tensile strength analysis, BNC membranes were completely dried at 80 °C in vacuum gel dryer (Model 543, Bio-Rad, Hercules, CA, USA) and cut into 2.0 cm wide strips. The experiment was carried out with parameters adjusted to 25 mm/min^−1^ of clamps movement velocity and 15 mm of initial distance between the clamps.

#### 4.5.2. Compression Tests

Wet BNC membranes, after seven-days-long stationary cultivation and purification, were cut into 2.5 × 2.5 cm squares and tested for compressive strength. The Zwick Roell testing machine was set at a compression speed of 25 mm/min^−1^ and a minimum distance between the clamps of 2 mm.

### 4.6. Bacteriostatic Properties Analysis

The antibacterial activity of the BNC-based composites was examined against both Gram-negative (*Escherichia coli*) and Gram-positive (*Bacillus subtilis*) bacteria using the agar disc diffusion method (Kirby–Bauer test). These representative strains were activated in 15 g/L of commercially available enriched broth (BTL, Łódź, Poland) and then incubated at 37 °C until an optical density at 600 nm (OD_600_) of 0.2 ± 0.02 ODU was reached. After the overnight cultivation of *E. coli* and *B. subtilis*, 0.01% of the bacterial suspension (~1.6 × 10^8^ CFU/mL) was transferred to sterile LB medium containing 1.5% agar (BTL, Łódź, Poland) and poured into the sterile Petri dishes. Meanwhile, the BNC-based composites in the wet state were cut into discs with a diameter 1 cm and sterilized at 121 °C for 15 min. Next, the samples were placed on top of previously poured Petri dishes with target bacterial suspensions, and subsequently incubated for 24 h at 37 °C. The bacteriostatic activity of the modified BNC membranes was monitored by observing inhibition zone formed around the discs. The experiment was performed in triplicate.

### 4.7. Statistical Analysis

Experimental data were expressed as mean ± standard deviation of at least three measurements and statistically processed using RStudio software (version 1.4.1106). To detect significant differences between bacterial nanocellulose synthesized in SH medium with and without ethanol, Student’s t tests were conducted. Statistical analyzes for multiple comparisons were performed using one-way analysis of variance (ANOVA), followed by the post hoc Tukey’s HSD procedure. In all experiments, the differences between groups were considered significant at a probability level of 0.05 (*p* ≤ 0.05). The assumptions of data normality and the homogeneity of variances were verified by Shapiro–Wilk and Bartlett tests, respectively.

## 5. Conclusions

Bacterial cellulose produced by different strains of the *Komagataeibacter* genus exhibits diverse physical, physico-chemical, and mechanical properties. Therefore, the appropriate selection of the BNC-synthesizing microorganisms is an essential step towards efficient, cost-effective, and targeted production of this biological material, especially with regard to its further functionalization and future applications. Since the comparative analysis and further microorganism choice is hard to perform, based on the outcomes provided by different sources and studies presented in the literature, it is highly important to deliver a more reliable source of information, i.e., investigate many BNC producing strains that are mentioned in the literature, as well as new prospective isolates. In this present study, a comparative analysis of bacterial cellulose membranes derived from chosen *Komagataeibacter* strains was carried out in terms of their specific properties, as well as further their potential for the fabrication of bio-inspired composites. The strain selection procedure, starting with evaluation of BNC efficiency, revealed that the greatest yields were obtained for the 1st and 4th strains and the variants numbered 3Et, 5Et, 6Et, 8Et, 9Et, 12Et, and 13Et with ethanol. These outcomes suggest that the chosen strains may be very promising from a financial point of view, but also due to the membranes properties, which can be seen in other parts of this study. The second step of bacterial screening revealed the microorganisms which produce the BNC films with the best mechanical strength parameters. The following six strains and cultivation variants were selected: 1, 4, 6Et, 8Et, 9Et, and 12Et. These membranes showed properties suitable for the formation of durable biological materials and nanocomposites with functional additives, which indicates their significant potential in various fields of industry and medicine. In the final selection step, the film’s compactness, structural integrity, overall appearance, and thickness were taken into account. Ultimately, in this study, three BNC-synthesizing strains (*K. hansenii* H3 (6Et), *K. rhaeticus* K4 (8Et), and *Komagataeibacter* sp., isolated from balsamic vinegar (12Et)), were selected as the most suitable producers for the further fabrication of bio-inspired nanocomposites. Bacterial cellulose was modified at the stage of its biosynthesis by supplementing the growth medium with five representative additives, such as citrus pectin (CP), apple pectin (AP), wheat starch (WS), polyvinyl alcohol (PVA), and polyethylene glycol (PEG). The selected polymeric compounds significantly changed the physico-chemical and mechanical properties of chemically inert cellulose, especially the dye loading (adsorption) abilities (BNC/CP, BNC/AP), tensile and compression strength (BNC/PVA, BNC/PEG), and water adsorption/retention capacities (BNC/PEG). Thus, the resulting modified biological materials can be potentially useful in various fields of industry and medicine and may become a valuable and profitable competitor against commercial materials currently available on the market.

## Figures and Tables

**Figure 1 ijms-23-03391-f001:**
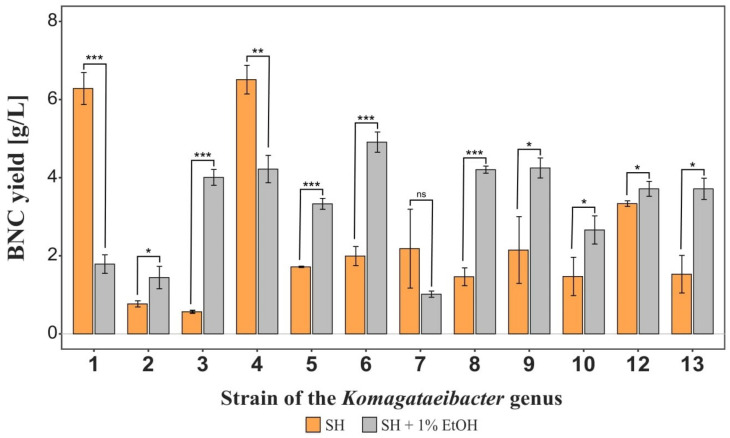
Cellulose biosynthesis yield by *Komagataeibacter* strains. Statistically significant differences were calculated using Student’s t test and shown as * = *p* ≤ 0.05, ** = *p* ≤ 0.01 and *** = *p* ≤ 0.001, compared to BNC produced without ethanol.

**Figure 2 ijms-23-03391-f002:**
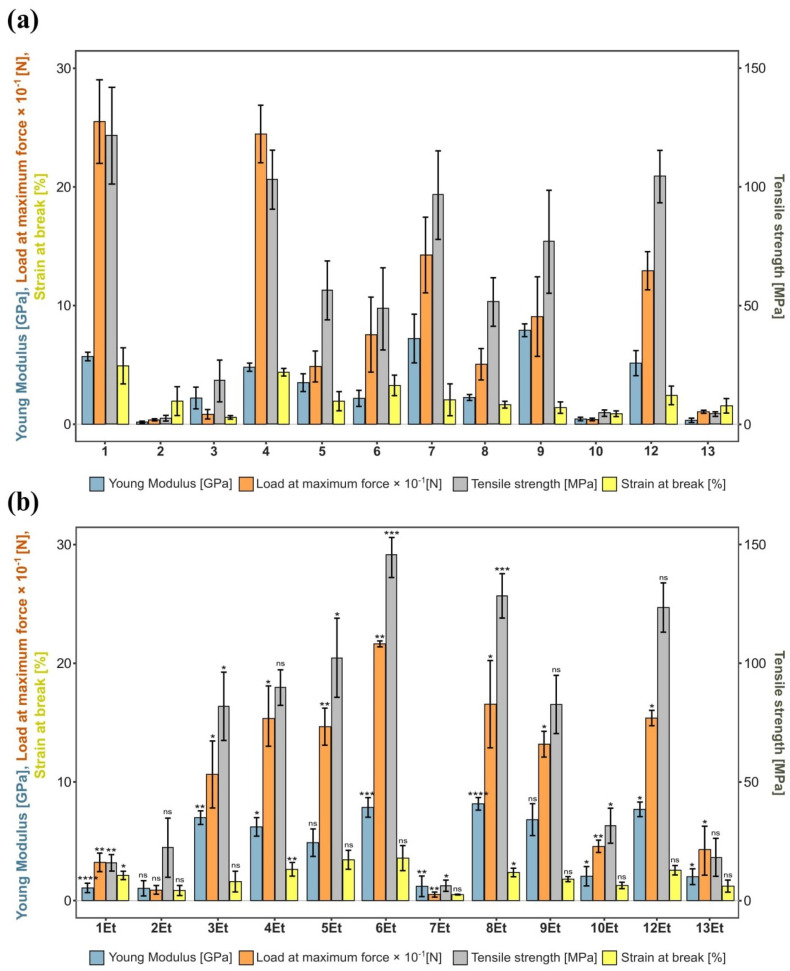
The results of biomechanical measurements when breaking bacterial cellulose membranes produced by 12 *Komagataeibacter* strains (**a**) without and (**b**) with ethanol supplementation. Statistically significant differences were calculated using Student’s t test and shown as * = *p* ≤ 0.05, ** = *p* ≤ 0.01, and *** = *p* ≤ 0.001, compared to BNC synthesized without ethanol.

**Figure 3 ijms-23-03391-f003:**
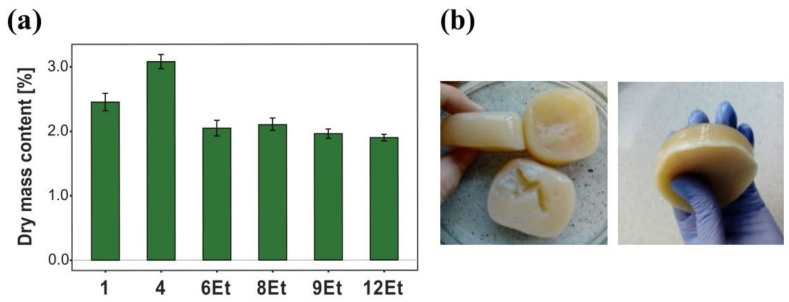
(**a**) Dry mass content in bacterial cellulose membranes synthesized by chosen strains, (**b**) more loose, gelatinous BNC synthesized by the 6th strain (*K. hansenii* H3).

**Figure 4 ijms-23-03391-f004:**
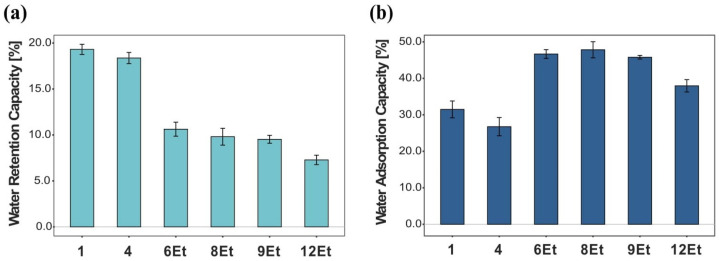
Comparison of the water related parameters obtained for BNC membranes produced by chosen strains. (**a**) Water retention capacity (WRC), (**b**) water adsorption capacity (WAC).

**Figure 5 ijms-23-03391-f005:**
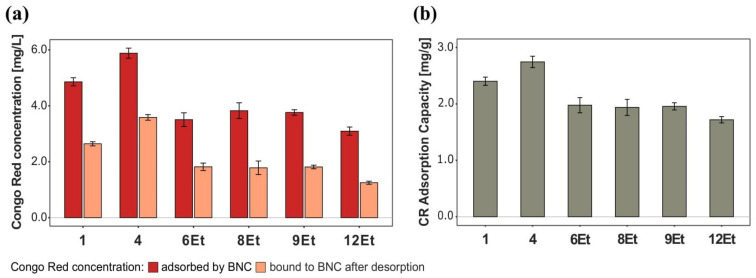
The results of Congo Red loading (adsorption-desorption) studies. (**a**) Comparison of the dye concentration adsorbed by BNC membranes to its quantity remaining in the material after rinsing process, (**b**) Congo Red adsorption capacity (CRC).

**Figure 6 ijms-23-03391-f006:**
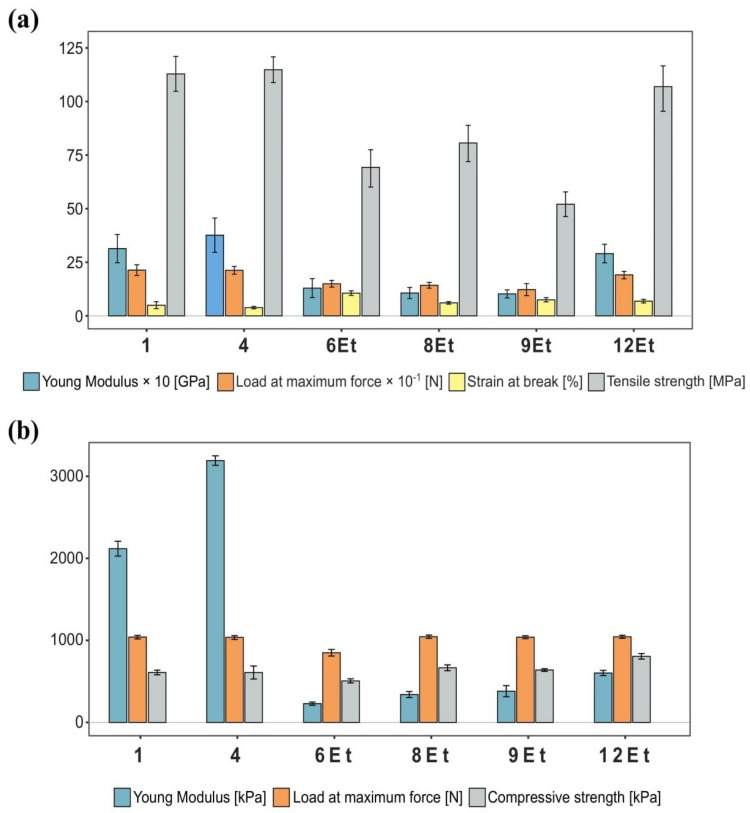
Mechanical parameters of BNC membranes synthesized by chosen strains. (**a**) Breaking strength measurements results, (**b**) compression tests results.

**Figure 7 ijms-23-03391-f007:**
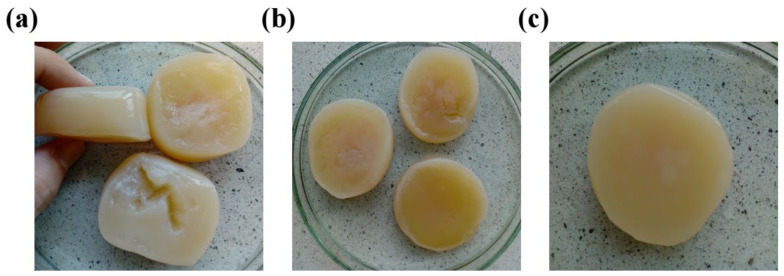
Bacterial cellulose membranes produced by chosen bacterial strains: (**a**) *K. hansenii* H3 (6Et), (**b**) *K. rhaeticus* K4 (8Et), and (**c**) *Komagataeibacter* sp. isolated from balsamic vinegar (12Et).

**Figure 8 ijms-23-03391-f008:**
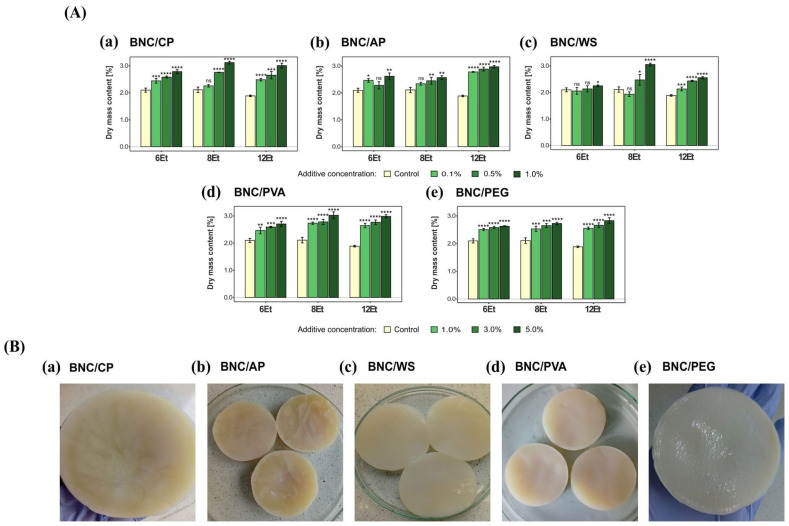
(**A**) Dry mass content in bacterial cellulose composites produced by *K. hansenii* H3 (6Et), *K. rhaeticus* K4 (8Et), and the strain isolated from balsamic vinegar (12Et). (**a**) BNC/CP, (**b**) BNC/AP, (**c**) BNC/WS, (**d**) BNC/PVA, and (**e**) BNC/PEG. Statistically significant differences were calculated using a one-way ANOVA with Tukey’s multiple comparison post-hoc test and are shown as * = *p* ≤ 0.05, ** = *p* ≤ 0.01, *** = *p* ≤ 0.001 and **** = *p* ≤ 0.0001, as compared to control (native) membranes. (**B**) Composites synthesized by the 6th strain: (**a**) BNC/CP and (**b**) BNC/AP with visible heterogenous streaks formed by pectins, (**c**) more loose, gelatinous BNC/WS films, (**d**) BNC/PVA and (**e**) BNC/PEG with compact, homogeneous structure and visible layer of added compounds.

**Figure 9 ijms-23-03391-f009:**
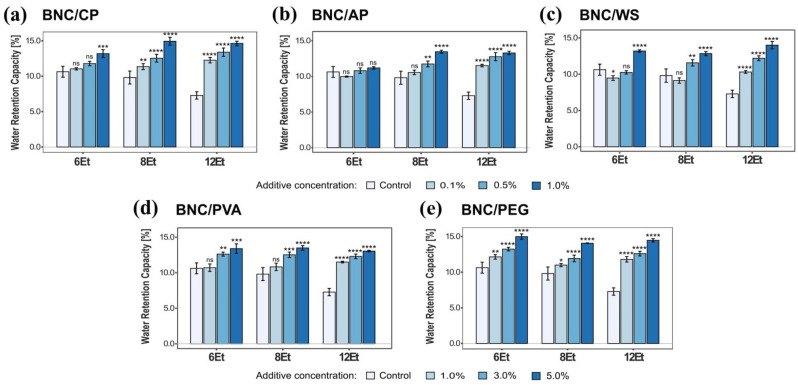
Water retention capacity of bacterial cellulose composites produced by *K. hansenii* H3 (6Et), *K. rhaeticus* K4 (8Et), and the strain isolated from balsamic vinegar (12Et). (**a**) BNC/CP, (**b**) BNC/AP, (**c**) BNC/WS, (**d**) BNC/PVA, and (**e**) BNC/PEG. Statistically significant differences were calculated using a one-way ANOVA with Tukey’s multiple comparison post-hoc test and are shown as * = *p* ≤ 0.05, ** = *p* ≤ 0.01, *** = *p* ≤ 0.001, and **** = *p* ≤ 0.0001, compared to control (native) membranes.

**Figure 10 ijms-23-03391-f010:**
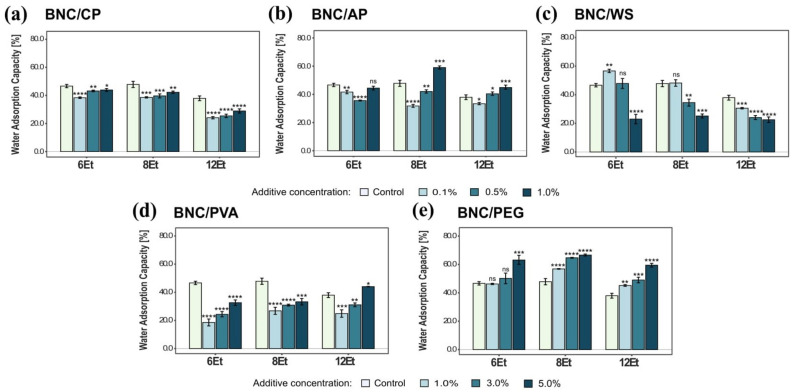
Water adsorption capacity of bacterial cellulose composites produced by *K. hansenii* H3 (6Et), *K. rhaeticus* K4 (8Et) and the strain isolated from balsamic vinegar (12Et). (**a**) BNC/CP, (**b**) BNC/AP, (**c**) BNC/WS, (**d**) BNC/PVA, and (**e**) BNC/PEG. Statistically significant differences were calculated using a one-way ANOVA with Tukey’s multiple comparison post-hoc test and are shown as * = *p* ≤ 0.05, ** = *p* ≤ 0.01, *** = *p* ≤ 0.001, and **** = *p* ≤ 0.0001, compared to control (native) membranes.

**Figure 11 ijms-23-03391-f011:**
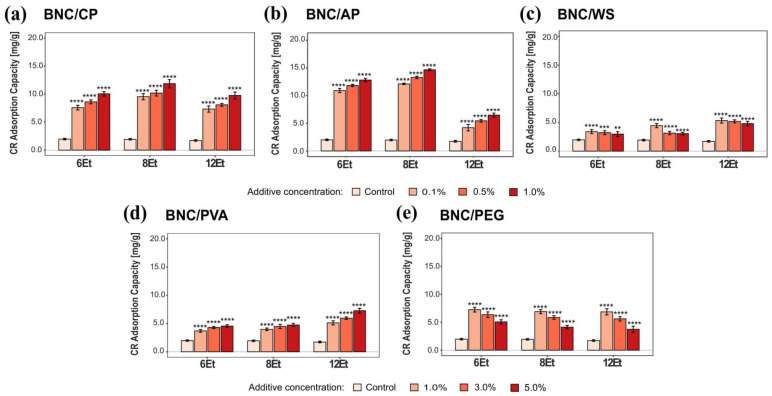
Congo Red adsorption capacity of bacterial cellulose composites produced by *K. hansenii* H3 (6Et), *K. rhaeticus* K4 (8Et), and the strain isolated from balsamic vinegar (12Et). (**a**) BNC/CP, (**b**) BNC/AP, (**c**) BNC/WS, (**d**) BNC/PVA, and (**e**) BNC/PEG. Statistically significant differences were calculated using a one-way ANOVA with Tukey’s multiple comparison post-hoc test and are shown as ** = *p* ≤ 0.01, *** = *p* ≤ 0.001, and **** = *p* ≤ 0.0001, compared to control (native) membranes.

**Figure 12 ijms-23-03391-f012:**
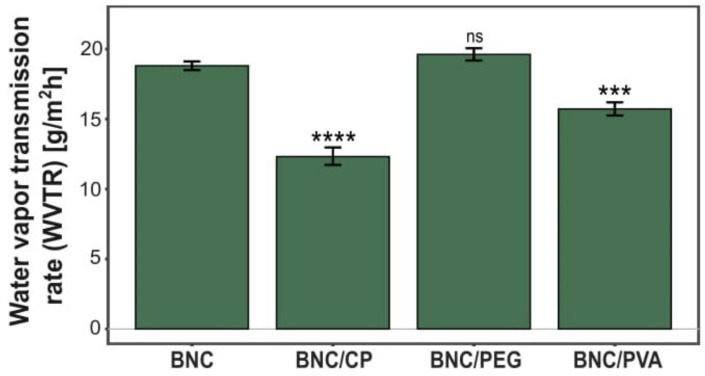
Water vapor transmission rate (WVTR) of native and modified BNC films produced by the strain isolated from balsamic vinegar (12Et). Statistically significant differences were calculated using a one-way ANOVA with Tukey’s multiple comparison post-hoc test and are shown as *** = *p* ≤ 0.001, and **** = *p* ≤ 0.0001, compared to control (native) membranes.

**Figure 13 ijms-23-03391-f013:**
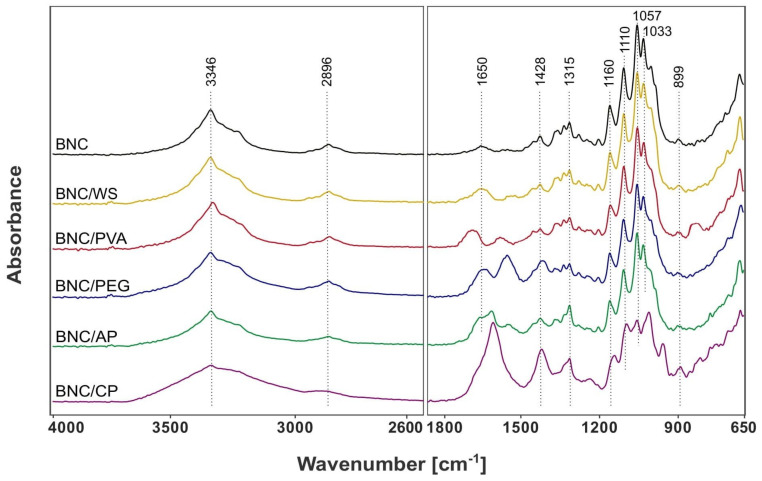
Comparison of ATR-FTIR spectra of native and modified BNC membranes.

**Figure 14 ijms-23-03391-f014:**
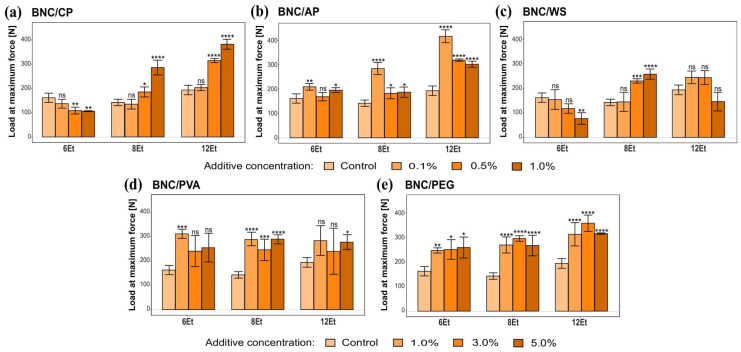
The maximum force required to break bacterial cellulose composites produced by *K. hansenii* H3 (6Et), *K. rhaeticus* K4 (8Et), and the strain isolated from balsamic vinegar (12Et). (**a**) BNC/CP, (**b**) BNC/AP, (**c**) BNC/WS, (**d**) BNC/PVA, and (**e**) BNC/PEG. Statistically significant differences were calculated using a one-way ANOVA with Tukey’s multiple comparison post-hoc test and are shown as * = *p* ≤ 0.05, ** = *p* ≤ 0.01, *** = *p* ≤ 0.001, and **** = *p* ≤ 0.0001, compared to control (native) membranes.

**Figure 15 ijms-23-03391-f015:**
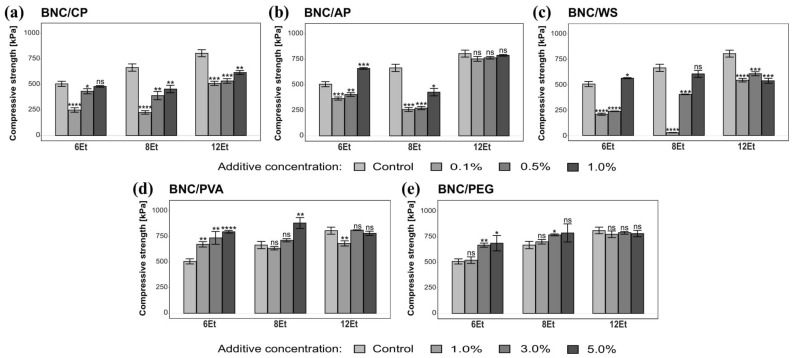
Compression strength of bacterial cellulose composites produced by *K. hansenii* H3 (6Et), *K. rhaeticus* K4 (8Et), and the strain isolated from balsamic vinegar (12Et). (**a**) BNC/CP, (**b**) BNC/AP, (**c**) BNC/WS, (**d**) BNC/PVA, and (**e**) BNC/PEG. Statistically significant differences were calculated using a one-way ANOVA with Tukey’s multiple comparison post-hoc test and are shown as * = *p* ≤ 0.05, ** = *p* ≤ 0.01, *** = *p* ≤ 0.001, and **** = *p* ≤ 0.0001, compared to control (native) membranes.

**Table 1 ijms-23-03391-t001:** Bacterial strains of the *Komagataeibacter* genus used in the experiments.

Strain Number	Systematic Name	Description
1	*K. hansenii* ATCC 53582	ATCC collection [39]
2	*K. hansenii* ATCC 23769	ATCC collection [72]
3	*K. xylinus* E25	Strain belonging to Bowil Biotech Ltd. [73]
4	*K. xylinus* E26	In-house collection of IMIB [12]
5	*K. xylinus* BCRC 12334	Received from Jyh Ming Wu, Department of Chemical and Materials Engineering, Chinese Culture University, Taipei, Taiwan [12]
6	*K. hansenii* H3	Received from Jose Fontana, Universidade Federal do Paraná (UFPR), Curitiba, Brazil
7	*K. hansenii* H4	Received from Jose Fontana, Universidade Federal do Paraná (UFPR), Curitiba, Brazil
8	*K. rhaeticus* K4	Isolated from Kombucha at IMIB
9	*K. rhaeticus* K3	Isolated from Kombucha at IMIB [24]
10	*K. hansenii* H5	Isolated from vinegar at IMIB
11	*A. xylinum* ŁOCK	Collection of Pure Cultures of the Institute of Microbiology and Fermentation, Lodz University of Technology [74]
12	*Komagataeibacter* sp.	Isolated from balsamic vinegar at IMIB
13	*K. hansenii* SI1	Isolated from Kombucha at IMIB [14]

**Table 2 ijms-23-03391-t002:** Summary of the conducted modifications of SH medium.

No.	Type of Additive	Concentration [% *w*/*v*]
1.	Citrus pectin (CP) + 12.5 mM CaCl_2_	0.1 0.5 1.0
2.	Apple pectin (AP) + 12.5 mM CaCl_2_	0.1 0.5 1.0
3.	Wheat starch (WS)	0.1 0.5 1.0
4.	Polyvinyl alcohol (PVA)	1.0 3.0 5.0
5.	Polyethylene glycol (PEG)	1.0 3.0 5.0

## Data Availability

The authors confirm that the data supporting the findings of this study are available within the article.

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
