# Peer review of "Comparative Analysis of Bacterial Cellulose Membranes Synthesized by Chosen Komagataeibacter Strains and Their Application Potential"

_ijms, 2022, doi:10.3390/ijms23063391_

Round 1

Reviewer 1 Report

In this manuscript, the author presented a comparative analysis of bacterial cellulose membranes synthesized by 13 strains of the Komagataeibacter genus. The BNC biosynthetic efficiency with and without the addition of ethanol was evaluated, followed by the assessment of the mechanical breaking strength and physical parameters (film’s compactness, structural integrity, appearance, and thickness) of the biological membranes. This is a data-intensive job, and the author has made a lot of efforts and conducted a comprehensive study, as can be seen from the abundant data. However, some issues I discovered while reading the manuscript were listed as follows:

Major:

  1. According to the exhibited data from Figure 1, the greatest yields of strains should be 3rd,4th,6 th,8 th, and 9 th (at L89). Similarly, the sample of 11th strain was contaminated as you said, and thus the following sentence “For the 5th, 11th and 12th ones the results were slightly lower” should be sentence “For the 5th, 12th and 13th ones the results were slightly lower” (at L89).
  2. At L101-111, I feel quite confused of the interpretation of result as the author’s description cannot match the data shown us (Figure 2). The author needs to rewrite and reanalyze this section carefully.

Minor:

  1. In terms of graphics formats, the horizontal lines in the middle of the Figures and Tables should not exist.
  2. At L217, “1040 N ± 5 N” should be “1040 ± 5 N”. Also, “stress-strain curves” should be “stress-strain curves” at L225.
  3. At L733, there is no space between numbers and units (“0.1g, 0.5g or 1.0g of citrus/apple pectin”). At L883, there is no unit in “0.2 ± 0.02”.

Author Response

Dear Sir of Madame,

We appreciate the kind review of our manuscript. We hope all the corrections we provided in the document will meet the expectations of Yours and will allow for the publication of the corrected article.

With kind regards,

Monika Kaczmarek

Reviewer 2 Report

Present manuscript deals with preparation of bacterial nanocellulose membranes using different bacteria strains and additives, with focus on their influence in biosynthesis yield and physico-chemical properties. In my opinion, it is an interesting study and should appeal to readers working in nanocellulose and related fields. Nevertheless, I believe the manuscript should be revised in some points before publication:

-Overall the manuscript contains to much data and text. Authors could try to move some of the content to supplementary files.

-Description of experimental procedures is well written, but in some cases it is not clear if tested membranes were dried or in hydrogel form when used for each type of characterization

-In my opinion, inclusion of SEM images is of crucial importance to understand influence of strain and additives on microstructure of prepared native membranes as well as composites ones. In this case I would suggest authors to carry out freeze drying or supercritical drying procedure to preserve porous structure of wet membranes.

-Dye adsorption capacity determination employed (single point measurement) is not  appropriate, authors should instead determine it by a adsorption isoterm.

Author Response

(The authors gave the same response as above.)

Round 2

Reviewer 1 Report

The authors have addressed the comments properly.

Author Response

Dear Sir of Madame,

We kindly thank you for your answer and feedback. We are glad that the corrections we provided in the manuscript met Your expectations.

With kind regards,

Monika Kaczmarek

Reviewer 2 Report

Authors provide suitable responses and edited manuscript accordingly. Thus I recommend manuscript for publication in the present form

Author Response

Dear Sir of Madame,

We kindly thank you for Your answer and feedback. We are glad that the corrections we provided in the manuscript met Your expectations.

With kind regards,

Monika Kaczmarek